# Factors Associated with Significant Weight Loss in Hospitalised Patients with COVID-19: A Retrospective Cohort Study in a Large Teaching Hospital

**DOI:** 10.3390/nu14194195

**Published:** 2022-10-08

**Authors:** Dimitra Zannidi, Pinal S. Patel, Eleni Leventea, Jessica Paciepnik, Frances Dobson, Caroline Heyes, Robert J. B. Goudie, Linda M. Oude Griep, Jacobus Preller, Lynsey N. Spillman

**Affiliations:** 1Nutrition and Dietetics Department, Addenbrooke’s Hospital, Cambridge University Hospitals NHS Trust, Cambridge CB2 0QQ, UK; 2MRC Biostatistics Unit, School of Clinical Medicine, East Forvie Building, University of Cambridge, Cambridge CB2 0SR, UK; 3NIHR Biomedical Research Centre, Diet, Anthropometry, and Physical Activity Group, MRC Epidemiology Unit, University of Cambridge, Cambridge CB2 0QQ, UK; 4Department of Acute Internal Medicine and Intensive Care, Cambridge University Hospitals NHS Foundation Trust, Cambridge CB2 0QQ, UK

**Keywords:** malnutrition, weight loss, COVID-19, nutritional risk, SARS-CoV-2, nutritional status

## Abstract

SARS-CoV-2 infection (COVID-19) is associated with malnutrition risk in hospitalised individuals. COVID-19 and malnutrition studies in large European cohorts are limited, and post-discharge dietary characteristics are understudied. This study aimed to assess the rates of and risk factors for ≥10% weight loss in inpatients with COVID-19, and the need for post-discharge dietetic support and the General Practitioner (GP) prescription of oral nutritional supplements, during the first COVID-19 wave in a large teaching hospital in the UK. Hospitalised adult patients admitted between March and June 2020 with a confirmed COVID-19 diagnosis were included in this retrospective cohort study. Demographic, anthropometric, clinical, biochemical, and nutritional parameters associated with ≥10% weight loss and post-discharge characteristics were described. Logistic regression models were used to identify risk factors for ≥10% weight loss and post-discharge requirements for ongoing dietetic input and oral nutritional supplement prescription. From the total 288 patients analysed (40% females, 72 years median age), 19% lost ≥ 10% of their admission weight. The length of hospital stay was a significant risk factor for ≥10% weight loss in multivariable analysis (OR 1.22; 95% CI 1.08–1.38; *p* = 0.001). In addition, ≥10% weight loss was positively associated with higher admission weight and malnutrition screening scores, dysphagia, ICU admission, and artificial nutrition needs. The need for more than one dietetic input after discharge was associated with older age and ≥10% weight loss during admission. A large proportion of patients admitted to the hospital with COVID-19 experienced significant weight loss during admission. Longer hospital stay is a risk factor for ≥10% weight loss, independent of disease severity, reinforcing the importance of repeated malnutrition screening and timely referral to dietetics.

## 1. Introduction

Malnutrition is a major public health problem, affecting more than 30% of hospitalised patients [1,2,3,4]. Malnutrition is associated with high mortality and morbidity, functional decline, prolonged hospital stay, and increased healthcare costs [2,5]. The co-existence of immobility and malnutrition with the disease processes that affect energy and protein balance and appetite can lead to progressive weight loss and sarcopenia [6,7].

SARS-CoV-2 infection (COVID-19) and related pneumonia have been significantly associated with malnutrition [8,9]. Patients with COVID-19 are at high risk for malnutrition due to an inflammatory syndrome associated with reduced food intake and increased muscle catabolism. Therefore, the prevention of malnutrition and nutritional management are the key aspects of care for patients with COVID-19 [10]. Previous studies have observed that patients with COVID-19 who have survived an intensive care admission often present with weight loss and respiratory, cardiovascular, and neurological impairments. These detrimental consequences require prolonged treatment which can further increase the risk of malnutrition [11,12]. The nutritional requirements of patients with COVID-19 are often not achieved due to anorexia and reduced olfactory senses, as well as nausea, vomiting, and diarrhoea which affect both dietary intake and nutrient absorption [13]. The pooled prevalence of malnutrition among hospitalised patients with COVID-19 was approximately 50% in a recent meta-analysis, with a higher prevalence among critically ill patients than general-ward patients (61.03% versus 48.19%, respectively). The odds of mortality among malnourished hospitalised patients with COVID-19 are also reported to be 10 times higher than the odds of mortality for those who were not malnourished [13].

Despite the aforementioned literature, little is known about COVID-19 hospitalised patients in European settings, as currently available data are mainly derived from small samples and short-duration studies and/or daily hospital audits, whilst the studies with larger samples are mainly conducted in Asian populations. Current COVID-19 nutrition-related literature focuses mostly on ICU and older populations, while the studies that include general-ward and ICU data and a wider age group range are limited. In addition, there is a gap in the evidence for the nutritional outcomes of patients with COVID-19 after hospital discharge, especially with reference to the need for dietetic input and oral nutritional supplement prescription.

The primary aim of this study was to identify the factors associated with significant weight loss (≥10% of initial weight) in inpatients with COVID-19 infection. The secondary aims were to identify the factors associated with the need for more than one contact for dietetic support and with the General Practitioner (GP) prescription of oral nutritional supplements (ONS) after discharge from the hospital.

## 2. Materials and Methods

### 2.1. Study Design and Setting

This is a retrospective observational study conducted at Cambridge University Hospitals (CUH), a tertiary National Health Service (NHS) University Hospital in England.

#### 2.1.1. Ethical Statement

The study was approved by a UK Health Research Authority ethics committee (20/WM/0125). The de-identified data presented here were collected during routine clinical practice; therefore, there was no requirement for informed consent.

#### 2.1.2. Study Population

All adults (≥18 years of age) admitted to CUH with COVID-19 from 01 March to 30 June 2020 were included in the study. The sample size was based on all eligible patients admitted to CUH within the first wave of COVID-19 in the UK. The diagnosis of COVID-19 was based on either a positive diagnostic SARS-CoV-2 test or a clinical diagnosis of COVID-19 during or up to 14 days prior to the hospital admission. Diagnostic testing used either a real-time reverse transcription polymerase chain reaction (RT-PCR) of the RdRp gene from a nasopharyngeal swab, or the SAMBA II point-of-care test used at the hospital [14]. All patients with clinically diagnosed COVID-19 (based on symptoms and in the clinical opinion of the treating clinician) were included, as diagnostic testing was limited during the early stages of the pandemic [15]. The clinical diagnosis of COVID-19 was identified using the International Classification of Diseases 10th Edition (ICD-10) codes [16] in the Electronic Health Records (EHRs). The following cases were excluded from the study: pregnant women, patients younger than 18 years old, and any day unit admissions and/or other admissions with a duration of less than 48 h.

#### 2.1.3. Data Collection

This study used routinely collected data extracted from the hospital’s electronic medical records. The primary and secondary outcomes of the study, as well as all independent variables, were decided a priori of data collection and analysis. Dietetic-related data were collected by a qualified clinical dietitian with research experience. All data regarding hospital admission and medical information, including admission details, blood test results, comorbidities, and medications, were provided by a statistician from the MRC Biostatistics Unit and the CUH clinical informatics team. We collected data from a single admission unless readmission occurred within 30 days of hospital discharge. The admission date was considered the patient’s first admission after or including a positive COVID-19 diagnosis. The discharge date was considered the last discharge date of all the admissions. Hospital stay duration was defined as the time between the first admission and final discharge and was reported both in days and weeks. ICU admission and the duration of ICU stay were also reported. Data collection at different time points was considered, including on/closest to hospital admission, on/closest to hospital discharge, and up to 6 weeks post-discharge. 

#### 2.1.4. Primary Outcome

The primary outcome was ≥10% weight loss during the hospital admission, as ≥10% weight loss is a National Institute for Health and Care Excellence (NICE) criterion for malnutrition and initiates the consideration of nutritional support providence [17]. A binary weight loss variable was created to define patients who experienced ≥10% weight loss and those with <10% weight loss.

#### 2.1.5. Secondary Outcomes

The secondary outcomes were defined as (1) more than one dietetic input after hospital discharge, including inputs from CUH dietitians or community dietitians, and (2) a request for an ONS prescription from the GP after hospital discharge.

#### 2.1.6. Demographics

Demographic data included age on admission, sex, and ethnicity. Ethnicity was grouped into a binary variable (white or other). 

#### 2.1.7. Anthropometry

Patients’ weight data were collected on hospital admission, on hospital discharge, and up to 6 weeks post-discharge. When weight was not recorded or estimated, the variable was reported as missing. The height information recorded in patients’ EHR was collected, and when discrepancies in the reported heights were observed, the one that was used in the dietetic notes was selected. BMI was calculated based on the available weight and height data and is presented in two different ways: a four-category variable based on the World Health Organization (WHO) criteria [18], and a binary variable for BMI < 25 kg/m^2^ and BMI ≥ 25 kg/m^2^. Weight loss was calculated as the difference between the weight on admission and the weight closest to hospital discharge.

#### 2.1.8. Malnutrition Risk on Admission

Malnutrition risk on admission (within 48 h from admission) was assessed by a validated nutritional screening score (NST) [19]. The NST score ranges between 0 and 14, with values 6 or above indicating high malnutrition risk and requiring dietetic referral.

#### 2.1.9. Disease-Related Factors

The following observations were included, either as recorded in EHRs or calculated: respiratory rate (RR); peripheral oxygen saturation on room air (SpO2); SpO2/FiO2 (S/F) ratio; Glasgow Coma Score (GCS); National Early Warning Score (NEWS) 2; serum C-reactive protein (CRP) on admission; and CRP max during admission. A GCS binary variable was created (GCS < 15 and GCS = 15) to identify the level of consciousness, with responsiveness to be equal with GCS = 15 and responsiveness impairments with GCS < 15. A ‘high’ CRP binary variable was created as CRP ≥ 178 mg/dL (the median max CRP value). COVID-19 and nutrition-related reported symptoms during admission were collected, including taste changes/loss, smell changes/loss, dysphagia/swallowing difficulties, and anorexia/poor appetite. The following comorbidities, based on the ICD-10 codes [16] reported in EHRs, were included in the analyses, as they have been identified as risk factors for COVID-19: obesity, type 2 diabetes mellitus (T2DM), hypertension, and chronic neurological conditions/disorders of the nervous system [20,21]. The treatment factors included oxygen therapy; ventilation (invasive mechanical ventilation, non-invasive mechanical ventilation, CPAP mask, high flow oxygen and humidified nasal oxygen); prone positioning; and renal replacement therapy (RRT).

The total number of deceased patients was reported. The mortality risk on admission was assessed using the ISARIC 4C Mortality Score tool [22]. The 4C Mortality Score is calculated using the total score of eight variables (age, sex, number of comorbidities, respiratory rate, peripheral oxygen saturation at room air, GCS, Urea, and CRP), and defines four risk groups with corresponding mortality rates determined as low risk (0–3 score, 1.2% mortality rate), intermediate risk (4–8 score, 9.9% mortality rate), high risk (9–14 score, 31.4% mortality rate), and very high risk (≥15 score, 61.5% mortality rate) [22]. Following the above categorisation, a categorical variable was created in order to assess the percentage of the study population within the four different risk groups.

#### 2.1.10. Dietary Management

Dietary management data were collected from the dietetic notes available on EHRs. The total number of dietetic inputs during admission was summed, including the total dietetic assessments and/or communications related to patient care. Route of nutrition included oral, nasogastric feeding (NG), nasojejunal feeding (NJ), percutaneous endoscopic gastrostomy/radiologically inserted gastrostomy (PEG/RIG), and parenteral nutrition (PN). The length of enteral tube feeding (EN) and/or PN was calculated including the days between the first report/day of EN/PN start and the last day of EN/PN provision. Unplanned breaks in EN/PN were excluded from the total days of artificial feeding provision. Oral nutrition supplement (ONS) provision and speech and language therapy (SLT) input data were collected.

#### 2.1.11. Post-Discharge

Post-discharge data were collected for up to 6 weeks after discharge. These included the data for secondary outcomes, the number of patients who received a dietetic call within 2 weeks post-discharge, total days between discharge and the dietetic call, discharge destination (usual residence, other/community hospital, nursing/residential home, etc.), feeding route on discharge, and patient-reported nutrition-related symptoms (taste changes, smell changes, dysphagia, and anorexia).

#### 2.1.12. Statistical Analysis

Descriptive characteristics are presented for categorical and continuous variables. Continuous parametric data are expressed as mean (±standard deviation), and continuous non-parametric data are expressed as median (25th–75th percentile). Categorical variables are summarised as frequencies and percentages. The comparisons between the groups (≥10% and <10% weight loss) were tested using Student’s *t*-test for continuous parametric variables, the Mann–Whitney test for non-parametric variables, and a Chi-squared (X²) test for categorical variables. All the tests were two-sided, and a *p*-value of <0.05 was considered statistically significant. Univariable and multivariable logistic regression analyses were used to estimate the adjusted odds ratios (ORs) for the primary and secondary outcomes with 95% confidence intervals (CIs). All univariable models were adjusted for age. All variables that emerged as risk factors (*p* < 0.05) at univariable analysis and did not demonstrate multicollinearity were used as covariates in the multivariable models. For the primary outcome (≥10% weight loss), two multiple regression models were created, with the second model including additional nutrition-related factors as markers of nutrition support (number of total dietetic contacts, ONS provision, and artificial feeding during admission). Age was retained in the models. Missing data were not imputed. Statistical analysis was conducted using IBM SPSS Statistics 27.0.

## 3. Results

A total of 423 patients with COVID-19 were admitted to the hospital between March and June 2020. From those, the data from 288 patients were included in the present analysis, as 135 patients had missing data regarding weight loss during admission. The descriptive characteristics of the analysed sample (n = 288) and the comparisons between the patients who lost <10% (n = 234) and ≥10% (n = 54) of their initial body weight during admission are presented in Table 1 and Table 2. The median weight on admission was 75.3 kg, and 57.0% of the included patients were classified within the overweight or obese category (BMI ≥ 25 kg/m^2^). In addition, 18.8% (n = 54) of the patients lost ≥10% of their initial body weight during their hospital admission. The frequency of patients who lost ≥ 10% of their weight was higher for those with a higher weight on admission. When the 4C Mortality Score was calculated, 23.6% (n = 25) of the total sample had intermediate risk, and 71.7% (n = 76) had high and very high mortality risk. Deceased patients (n = 56) were excluded from the post-discharge analyses. The sample’s post-discharge characteristics are described in Table 3.

### 3.1. Primary Outcome

The univariable risk factors for ≥10% weight loss are shown in Appendix A. The results of the multivariable regression for ≥10% weight loss are presented in Table 4. Only the length of stay (OR expressed per 1 week: 1.22; 95% CI 1.08–1.38, *p* = 0.001) was identified as an independent risk factor of ≥10% weight loss. The length of stay remained the only independent risk factor of ≥10% weight loss (OR 1.32; 95% CI 1.09–1.61, *p* = 0.005) in the second multivariable model which included additional nutrition-related factors (the number of total dietetic contacts, ONS provision, and artificial feeding during admission) (Appendix A).

### 3.2. Secondary Outcomes

Of those patients discharged from the hospital, 15% (n = 35) required more than one dietetic input after hospital discharge, and 12% (n = 27) required a request for ONS on prescription from the GP. The univariable logistic regression results for the secondary outcomes are presented as supplementary data (Appendix A). The results from multivariable logistic regression for patients who required more than one dietetic input after hospital discharge are shown in Table 5. The statistically significant, independent predictors for more than one dietetic input post-discharge were age (OR 1.03; 95% CI 1.00–1.05, *p* = 0.024) and ≥10% weight loss during admission (OR 2.11; 95% CI 1.03–4.34, *p* = 0.041) (Table 5). No statistically significant associations were found in univariable analyses for the secondary outcome, i.e., the request for GP ONS prescription (Supplementary Table 4).

## 4. Discussion

The primary aim of this study was to identify the risk factors for ≥10% weight loss in patients with COVID-19 during hospitalisation, both in the ICU and general-ward settings, in a large teaching hospital during the first wave of the COVID-19 pandemic. Notably, ≥10% weight loss of body weight has been previously associated with malnutrition in recovering patients with COVID-19 with a history of hospitalisation [10,23,24,25], with 9.6–18.0% reported as the incidence range of in-hospital malnutrition, in the studies that used similar definitions of malnutrition [24,26]. Of the study sample, 18.8% (n = 54) experienced ≥10% weight loss and were considered at high risk of in-hospital malnutrition.

Furthermore, ≥10% weight loss during hospital admission was positively associated with the length of hospital stay. These results are consistent with previously published reports, showing the length of hospital stay to be a significant risk factor for malnutrition [24,27]. The length of stay does not precede weight loss and is likely to be a marker of disease severity; however, high CRP, SpO2/FiO2 ratio, and ICU admission (proxy markers of disease severity) did not remain significant in the multivariable analysis, suggesting that a longer length of stay may increase the risk of significant weight loss, independent of disease severity. Even short periods of bed rest can induce reductions in muscle protein synthesis, resulting in the loss of muscle mass and strength [28]. Although the current study did not measure body composition, it is likely that part of the weight loss observed was the loss of lean body mass following the period of bed rest [29,30], which reinforces the importance of repeated hospital screening for malnutrition risk.

Compared with the <10% weight loss group, those with ≥10% weight loss had markers for worse disease severity including higher CRP max during admission, lower GCS on admission, and higher rates of oxygen therapy, ICU admission, and mechanical ventilation. Smell changes and dysphagia were also more common. These factors may be important indicators for dietetic support and may present before some nutrition screening tools identify malnutrition risk. The admission NST score was significantly higher in the ≥10% weight loss group, indicating a high risk of malnutrition on admission and reinforcing the current guidelines that recommend timely malnutrition screening and the need for dietetic referral to facilitate adequate nutritional support [31]. Despite the previous findings in the literature [13,20,21], our study did not show any differences in the prevalence of T2DM or hypertension, between the <10% and ≥10% weight loss groups; however, we found that the prevalence of chronic neurological conditions was higher in the ≥10% weight loss group. The presence of chronic neurological conditions has been previously associated with malnutrition [32] and was an independent predictor of mortality in hospitalised patients with COVID-19 [33]. However, to our knowledge, there are no previously published data that have associated chronic neurological conditions and malnutrition in patients with COVID-19. Furthermore, the patients in the ≥10% weight loss group received more dietetic input and a longer duration of artificial feeding, suggesting that dietetic care was appropriately directed to the patients who required it most. However, despite this, significant weight loss was experienced, indicating the severe nutritional consequences of COVID-19 infection requiring hospitalisation. Early nutritional support in hospitalised patients at malnutrition risk has been shown to significantly decrease 30-day complications and mortality in comparison to patients on a standard hospital diet [34]. 

Post-discharge, a significant proportion of patients required more than one dietetic contact for ongoing nutrition support (15%) and ONS prescription from the GP (12%). Older age and ≥10% weight loss during admission were positively associated with the need for ongoing dietetic input. These are important findings that indicate this patient population requires continued dietetic support and reinforces the need for dietetic workforce planning and primary care planning for ONS. The rehabilitation pathways for COVID-19 recovery suggest including screening for malnutrition, care plans for nutrition support, and the continuity of nutritional care between settings [31]; however, there is a paucity of evidence of the long-term need for post-discharge nutrition support and its effect in those patients previously hospitalised with COVID-19. 

This study’s strengths include a long study period, a large and varied sample of all adult COVID-19 inpatients, both in general-ward and ICU settings, and taking into account both the admission and post-discharge data. Malnutrition research in patients with COVID-19 has focused on the ICU settings; our study includes n = 192 hospitalised patients who did not require ICU admission, providing knowledge of malnutrition-related factors for non-ICU hospitalised patients. In our study, all the nutrition-related data were extracted from EHRs by a qualified dietitian, allowing for an accurate interpretation and exclusion of estimated and/or carried-over admission and discharge weights. Similarly, we had access to the expertise of a clinical statistician to identify other medical/admission-related data. This allowed us the benefit of a varied dataset both in clinical and dietetic-related parameters, assisting with the creation and analysis of comprehensive multivariable models.

One limitation is the retrospective study design, which does not allow us to investigate causality. Additionally, the lower rates of some symptoms observed in this study are attributed to the patients not being specifically asked about their symptoms, as the data collection was solely dependent on those symptoms reported in EHRs. Weight data were missing or inconsistent for 31% (n = 135) of the total patients admitted to the hospital (n = 432), following the challenges the healthcare system presented during the first COVID-19 wave. We did not present the data for the missing cases. However, the differences in characteristics between the patients with and without missing weight data were explored. A higher proportion of those with missing data died (38% vs. 19%) or had a shorter length of stay (6 vs. 17 days), and for these patients, we would expect there to be limited repeated measured weights during admission and, therefore, missing data. When data are missing not at random, as with this research, a complete case analysis is recommended over multiple imputations, as described by Hughes et al. [35]. Another possible limitation is that a large proportion (around 75%) of the sample was >70 years old which although is an accurate representation of the hospital admission population, may affect the results. We, therefore, adjusted for age in all univariate and multivariate regression analyses. Lastly, the change in dietetic referral criteria during the first wave of COVID-19 to include all patients admitted with COVID-19, irrespective of their malnutrition risk, may have limited the frequency of the NST used by the ward staff. This possibly explains the high rates of patients seen by the dietitians which may have minimised weight loss for some participants.

## 5. Conclusions

In conclusion, this study observed a 19% incidence of ≥10% weight loss during hospital stay, with the length of hospital stay as a significant, independent risk factor. A wide range of medical and nutrition-related factors were associated with in-hospital malnutrition in patients with COVID-19, and the findings reinforce the need for early nutritional screening and appropriate nutritional assessment and intervention, as previously highlighted by ESPEN [36]. Our findings suggest that the length of stay could be incorporated as an important criterion in nutritional screening tools. Furthermore, repeated malnutrition screening during hospital stays to identify weight loss and initiate a timely referral to the dietitian, education about the prevalence and impact of malnutrition, and guidance about potential additional risk factors should be considered and included in Trusts’ policies and procedures. Post-discharge pathways to incorporate ongoing dietetic input are also required. Longitudinal studies are needed to assess the role of nutritional input in the long-term prognosis of the disease and further assist in creating effective nutritional management strategies to improve the nutritional status of hospitalised patients with COVID-19.

## Figures and Tables

**Table 1 nutrients-14-04195-t001:** Descriptive characteristics of the total sample (n = 288) and comparisons between patients with ≥10% weight loss (n = 54) versus <10% weight loss during admission (n = 234).

Variables	Total Sample (n = 288)	Weight Loss <10% (n = 234)	Weight Loss ≥10% (n = 54)	*p* Value
Demographics				
Age, years	72.0 (59.0–82.0)	72.0 (61.0–82.0)	69.5 (56.0–79.2)	0.319
Sex, % (n.)				0.144
Male	59.7 (172)	57.7 (135)	68.5 (37)	
Female	40.3 (116)	42.3 (99)	31.5 (17)	
Ethnicity, % (n.)				0.554
White	75.3 (217)	76.1 (178)	72.2 (39)	
Other	24.7 (71)	22.2 (52)	27.8 (15)	
Comorbidities				
Chronic neurological conditions, % (n.)	29.9 (86)	**26.9 (63)**	**42.6 (23)**	**0.023**
T2DM, % (n.)	26.0 (75)	25.6 (60)	27.8 (15)	0.095
Hypertension, % (n.)	52.8 (152)	53.8 (126)	48.1 (26)	0.450
Admission data				
GCS on admission	15 (14–15)	**15 (14–15)**	**14 (5–15)**	**0.001**
NEWS2 on admission	5 (2–7)	4 (2–6)	5 (2–8)	0.149
SpO2/FiO2 ratio on admission	447.6 (323.2–461.9)	450.0 (342.8–462.0)	442.8 (285.4–457.1)	0.081
SpO2 on admission, %	96 (94–97)	96 (94–97)	95 (93–97)	0.687
Respiratory rate on admission (breaths/min)	20 (18–26)	20 (18–26)	20 (17–24)	0.257
CRP on admission, mg/dL	76.0 (29.0–175.0)	76.0 (25.0–177.0)	81.0 (38.5–172.5)	0.406
During admission data				
CRP max, mg/dL	191.0 (90.2–280.5)	**180.5 (79–286)**	**228.5 (158–305)**	**0.018**
CRP ≥ 178 mg/dl, % (n.)	53.5 (153)	**50.0 (117)**	**66.7 (36)**	**0.031**
Oxygen therapy, % (n.)				
Invasive mechanical ventilation	26.0 (75)	**21.8 (51)**	**44.4 (24)**	**0.001**
NIV	15.3 (44)	**12.4 (29)**	**27.8 (15)**	**0.005**
CPAP	14.2 (41)	**11.5 (27)**	**25.9 (14)**	**0.006**
Prone positioning	12.8 (37)	11.5 (27)	18.5 (10)	0.167
HFNO	7.6 (22)	7.3 (17)	9.3 (5)	0.619
RRT, % (n.)	10.8 (31)	**7.7 (18)**	**24.1 (13)**	**<0.001**
Length of hospital stay (days)	16.9 (10.9–30.9)	**15.5 (6.4–24.0)**	**35.8 (22.7–57.5)**	**<0.001**
Admitted to ICU, % (n.)	33.3 (96)	**28.2 (66)**	**55.6 (30)**	**<0.001**
Length of ICU stay (days)	14.9 (3.1–24.9)	12.6 (4.9–21.0)	17.9 (7.7–32.0)	0.068
Deceased, % (n.)	19.4 (56)	21.4 (50)	11.1 (6)	0.086
Readmitted within 30 days, % (n.)	18.4 (53)	18.8 (44)	16.7 (9)	0.715

Results are expressed as median (25th–75th percentile) for continuous data and % (n.) for categorical data. Bold font: statistically significant results (*p* < 0.05). Abbreviations: CPAP: continuous positive airway pressure, CRP: C-reactive protein, GCS: Glasgow Coma Scale, HFNO: high-flow nasal oxygen, ICU: intensive care unit, NEWS2: National Early Warning Score 2, CPAP: continuous positive airway pressure, NIV: non-invasive ventilation, RRT: renal replacement therapy, SpO2: peripheral oxygen saturation, T2DM: type 2 diabetes mellitus.

**Table 2 nutrients-14-04195-t002:** Nutrition-related characteristics of the analysed sample (n = 288) and comparisons between patients with ≥10% weight loss (n = 54) versus <10% weight loss during admission (n = 234).

Variables	Total Sample (n = 288)	Weight Loss <10% (n = 234)	Weight Loss ≥ 10% (n = 54)	*p* Value
Admission data				
Weight (kg)	75.3 (62.9–90)	**73.8 (62.3–88.2)**	**80.1 (65.5–98.1)**	**0.014**
BMI category, % (n.)				0.122
Underweight, BMI < 18.5 kg/m^2^	5.6 (16)	6.4 (15)	1.9 (1)	
Normal weight, BMI 18.5–24.9 kg/m^2^	37.5 (108)	39.3 (92)	29.6 (16)	
Overweight, BMI 25–29.9 kg/m^2^	27.8 (80)	27.8 (65)	27.8 (15)	
Obese, BMI ≥ 30 kg/m^2^	29.2 (84)	26.5 (62)	40.7 (22)	
Malnutrition screening tool score ≥ 6, % (n.)	26.7 (77)	**23.5 (55)**	**40.7 (22)**	**0.012**
During admission data				
Taste changes/loss, % (n.)	14.9 (43)	13.2 (31)	22.2 (12)	0.095
Smell changes/loss, % (n.)	7.6 (22)	**5.6 (13)**	**16.7 (9)**	**0.006**
Dysphagia, % (n.)	29.2 (84)	**25.2 (59)**	**46.3 (25)**	**0.002**
Anorexia/loss of appetite, % (n.)	64.9 (187)	64.1 (150)	68.5 (37)	0.540
Seen by dietitian, % (n.)	81.6 (235)	**78.6 (184)**	**94.4 (51)**	**0.007**
Number of total dietetic inputs	5 (2–9)	**5 (2–8)**	**8 (4–19)**	**<0.001**
Artificial feeding (EN, PN), % (n.)	32.6 (94)	**28.6 (67)**	**53.7 (29)**	**<0.001**
Duration of EN (days)	25.0 (12.0–37.5)	**17.0 (10.7–30.0)**	**36.0 (30.0–63.0)**	**<0.001**
Duration of PN in days	5.0 (2.0–12.5)	5.00 (2.0–12.0)	5.00 (2.0–28.0)	0.817
ONS provided, % (n.)	64.9 (187)	62.8 (147)	74.1 (40)	0.118
SLT assessment, % (n.)	30.6 (88)	**24.4 (57)**	**57.4 (31)**	**<0.001**

Results are expressed as median (25th-75th percentile) for continuous data and % (n.) for categorical data. Bold font: statistically significant results (*p* < 0.05). Abbreviations: BMI: body mass index, ONS: oral nutritional supplement, SLT: speech and language therapist, EN: enteral nutrition, PN: parenteral nutrition.

**Table 3 nutrients-14-04195-t003:** Post-discharge descriptive characteristics of the total sample (n = 232).

Variables	Post-Discharge Sample (n = 232)
Feeding route on discharge, % (n.)	
Oral	94.4 (219)
AF	8.6 (20)
Discharge destination, % (n.)	
Usual place of residence	74.1 (172)
Private/NHS nursing/Residential home	13.3 (31)
NHS hospital	8.6 (20)
Other	3.9 (9)
Received a dietetic call post-discharge, % (n.)	50.4 (117)
Duration between discharge and dietetic call (days)	9.0 (5.7–13.2)
More than one dietetic input after discharge, % (n)	15.1 (35)
Referred to community or HEF dietitians, % (n.)	13.4 (31)
ONS GP prescription requested, % (n.)	11.6 (27)
Weight post-discharge (kg)	72.4 (61.8–83.1)
Taste changes/loss, % (n.)	7.8 (18)
Smell changes/loss, % (n.)	3.4 (8)
Dysphagia, % (n.)	11.2 (26)
Anorexia/loss of appetite, % (n.)	15.1 (35)

Results are expressed as median (25th–75th percentile) for continuous data and % (n.) for categorical data. Abbreviations: AF: artificial feeding, CUH: Cambridge University Hospitals, HEF: home enteral feeding, ONS: oral nutritional supplement.

**Table 4 nutrients-14-04195-t004:** Multivariable binary logistic regression for risk factors of ≥10% weight loss during admission.

	≥10% Weight Loss during Admission	
Variables	OR (95% CI)	*p* Value
Age (years)	1.02 (0.99–1.04)	0.233
Sex (male)	1.32 (0.63–2.74)	0.462
Length of hospital stay (weeks)	**1.** **22 (1.08–1.38)**	**0.001**
ICU admission	1.51 (0.59–3.87)	0.389
CRP ≥178 mg/dl during admission	1.15 (0.55–2.40)	0.717
Weight on admission (kg)	1.04 (0.96–1.14)	0.323
Dysphagia on/during admission	0.98 (0.44–2.13)	0.931
SpO2/FiO2 ratio on admission	0.99 (0.99–1.00)	0.553
Chronic neurological conditions	1.02 (0.48–2.19)	0.959
Malnutrition screening tool score ≥6	1.46 (0.66–3.24)	0.346

Total number of patients included in the analysis n = 275, ≥10% weight loss during admission n = 52, <10% weight loss during admission n = 223. OR: odds ratio, 95% CI: 95% confidence interval. Bold font: statistically significant results (*p* < 0.05). Abbreviations: CRP: C-reactive protein, ICU: intensive care unit. Comorbidity chronic neurological conditions include ICD-10 codes G00-G99: Diseases of the nervous system. OR for weight on admission is expressed for every 5 kg of weight increase; OR for length of hospital stay is expressed per 1 week.

**Table 5 nutrients-14-04195-t005:** Multivariable logistic regression for patients who required more than one dietetic input after hospital discharge.

	Required More Than One Dietetic Input Post-Discharge	
Variables	OR (95% CI)	*p* Value
**Age (years)**	**1.03 (1.00–1.05)**	**0.024**
Sex (male)	1.45 (0.73–2.85)	0.287
CRP ≥178 mg/dl during admission	1.51 (0.77–2.97)	0.227
**≥10% Weight loss during admission**	**2.11 (1.03–4.34)**	**0.041**
Artificial feeding during admission	1.49 (0.71–3.12)	0.290

Total patients included in the analysis n = 230; required more than one dietetic input post-discharge n = 57; did not require more than one dietetic input post-discharge n = 173. OR: odds ratio, 95% CI: 95% confidence interval. Bold font: statistically significant results (*p* < 0.05). Abbreviations: CRP: C-reactive protein.

## Data Availability

The data that support the findings of this study are available from the corresponding author, but restrictions apply to the availability of these data, which were used under licence for the current study and, therefore, are not publicly available. The data are, however, available from the authors upon reasonable request subject to permission being obtained from Cambridge University Hospitals. For the purpose of open access, the author has applied a Creative Commons Attribution (CC BY) licence to any Author Accepted Manuscript version arising.

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
