# Peer review of "Factors Associated with Significant Weight Loss in Hospitalised Patients with COVID-19: A Retrospective Cohort Study in a Large Teaching Hospital"

_nutrients, 2022, doi:10.3390/nu14194195_

Round 1

Reviewer 1 Report

This manuscript addresses an important of dietetic support among hospitalised patients with COVID-19.

I have just a few comments, mainly on Materials and Methods section as shown below.

1. Exposure must be prior in time to the result. Does the observation of the risk factor precede the observation of 10% or more body weight?

2. How about P<0.2 as a variable selection for logistic regression, which may be entered with a wider range?

Reviewer 2 Report

In my opinion the manuscript entitled „Factors Associated with Significant Weight Loss in Hospitalised Patients with COVID-19: A Retrospective Cohort Study in a Large Teaching Hospital.” presents an interesting results regarding risk factors for significant weight loss in patients with COVID-19 patients, but I have some comments on this study and how the data is analysed.

My comments:

v  Authors included patients with COVID-19 and other diseases in this study. Taking into account that there are diseases that cause malnutrition or aggravate it, please explain why the study group was not narrowed down. Did the study participants only have the three (Chronic neurological conditions, T2DM, Hypertension) conditions listed in the table1?

v  It is known that the risk of malnutrition is influenced not only by disease but also by age. Please explain the inclusion of 18-year-old patients in the study, especially since the average in the group was over 70 years old. I suggest analysing the influence of age, possibly the division of the group by age.

v  How was the sample for the study calculated?

v  The number of enrolled participants is 423 and the final number of patients is 316, please remove from the analyses the cases with missing data so that the number is the same.

v  Line 268-283 - this paragraph is a repetition of the results, please refer your own results to the results obtained by other authors.

v  Please indicate how the obtained results can be used in a practical way.

Round 2

Reviewer 2 Report

Authors revised the manuscript according with the comments.